# Reflected generalized concentration addition and Bayesian hierarchical models to improve chemical mixture prediction

Daniel Zilber⬛, Kyle Messier⬛*

Division of Translational Toxicology, National Institute of Environmental Health Sciences, Durham, NC, United States of America

* kyle.messier@nih.gov

**Data Availability Statement:** All code written in support of this publication is publicly available at https://github.com/Spatiotemporal-Exposures-and-Toxicology/RGCA-DP. The Tox21 data is available at https://tripod.nih.gov//tox21/pubdata/.

## Abstract

Environmental toxicants overwhelmingly occur together as mixtures. The variety of possible chemical interactions makes it difficult to predict the danger of the mixture. In this work, we propose the novel Reflected Generalized Concentration Addition (RGCA), a piece-wise, geometric technique for sigmoidal dose-responsed inverse functions that extends the use of generalized concentration addition (GCA) for 3+ parameter models. Since experimental tests of all relevant mixtures is costly and intractable, we rely only on the individual chemical dose responses. Additionally, RGCA enhances the classical two-step model for the cumulative effects of mixtures, which assumes a combination of GCA and independent action (IA). We explore how various clustering methods can dramatically improve predictions. We compare our technique to the IA, CA, and GCA models and show in a simulation study that the two-step approach performs well under a variety of true models. We then apply our method to a challenging data set of individual chemical and mixture responses where the target is an androgen receptor (Tox21 AR-luc). Our results show significantly improved predictions for larger mixtures. Our work complements ongoing efforts to predict environmental exposure to various chemicals and offers a starting point for combining different exposure predictions to quantify a total risk to health.

## 1 Introduction

For many years, the field of toxicology focused on understanding how individual chemicals affect biological systems. However, it has become clear in recent years that individual chemicals can have a cumulative effect: exposure to safe levels of multiple chemicals can be toxic in aggregate [1, 2]. Even worse, durable "forever" chemicals accumulate in the environment and contribute to the growing list of chemicals people are exposed to every day [3]. Hence, being able to predict the toxicity of a mixture has moved from being an esoteric toxicological puzzle to an urgent challenge to understand looming health risks.

Mixture prediction is mainly a logistical problem for standard toxicology methods. For example, for a set of 10 chemicals, there are more than a thousand combinations of drugs, and for each of those combinations there are countless ways to set the relative concentrations. How

**Funding:** This work is supported by the National Institute of Environmental Health Sciences, Division of Translational Toxicology, Division of Intramural Research, and the Spatiotemporal Exposures and Toxicology group under project number ZIA ES103368-02. The funders had no role in study design, data collection and analysis, decision to publish, or preparation of the manuscript.

**Competing interests:** The authors have declared that no competing interests exists.

does one go about collecting all that data at reasonable cost, and what is to be done if a new chemical is added to the list? Realistically, there are thousands of chemicals that could potentially contribute to a mixture, so an experimental approach is infeasible. Recent spatial exposure maps such as those created in [4] also illustrate the variety of mixtures that people across the country are exposed to and present a compelling use-case for a predictive model. An observational study may seem like a natural work-around to the cost of experiments, but data collection is still tricky and multicollinearity makes the analysis difficult [5]. For these reasons, we focus on methodology that can predict a mixture effect directly from the individual chemical properties rather than a regression approach which requires experimental observations to fit a model.

The two most popular classical methods for predicting a mixture are concentration addition [6] and independent action [7]. Concentration addition (CA) models a mixture of chemicals as if they were different amounts of the same molecule; concentrations are scaled relative to their potency and added together as the name implies. The biological assumption is that all mixture components have the same mode of action. In contrast, independent action (IA) [8] assumes all modes of action are different, so a series of chemical responses can be added after an appropriate relative scaling. This is response addition, in contrast to the doses being scaled and added as in CA. Both models assume the chemicals are additive and do not synergize, a shortcoming that is not addressed in this work.

Often, neither CA nor IA is accurate [7], and a number of methods offer a middle path. Two-step or joint models such as those used in [9–11] apply CA to chemicals assumed to have similar modes of action and IA to combine the effects from different modes of action. Mode of action is not always known and quantitative structure-activity relationships (QSAR) can be used to inform the joint procedure [12, 13], but the relevant QSARs have to be selected either by an expert or through a procedure fitted to an observed mixture response. When the mixture response is available, techniques such as fuzzy membership models [14, 15] and integrated two-step models [16] can also be applied, but as mentioned we wish to avoid the implied experimental constraint.

When applying a two-step method, it is desirable to use the most accurate models for the individual dose responses. CA requires all chemical to have equal maximum effects. Generalized concentration addition (GCA) is a powerful extension of CA [17] that can account for partial agonists, or chemicals that are less effective than others and serve to antagonize large effects if present at high concentrations. GCA requires strong simplifying assumptions to model individual dose effects, so [18] present an alternative to GCA that allows for more complex models by fixing the toxic effect of partial agonists at higher concentrations. While they have reasonably good empirical results, their algorithm is essentially making the same compromise as GCA: the true sill parameter is sacrificed to correctly model the slope parameter that is ignored in GCA.

We propose a new technique that provides uncertainty quantification and generalizes the approach of GCA when the individual toxic responses are modeled with a three-parameter Hill or logistic function. The parameters represent a maximum effect, a dose to reach 50% of the maximum effect, and a slope, and our method allows for all three parameters to vary in contrast to the fixed slope value of 1 required by GCA. A key observation is that the standard case with unit slope exhibits symmetry that can be applied in the general case by defining an inverse based on reflection in a piecewise fashion. Like GCA, there is an option to use this inverse within a two-step framework where chemicals can be grouped together. Chemicals within a group are combined into a single effective response with GCA and then the responses are combined with IA.

The rest of this paper is organized as follows. In section 2 we describe the details of our method, including the Bayesian model, the reflection argument for extending the inverse, and possible implementations of the two-step model with uncertainty quantification. In section 2.5 we describe the motivating mixtures problem with data from the Tox21 database [19]. In section 3 we describe the results of our method both on simulated data and on the real data, including diagnostics for the parameter estimation procedures. Section 4 concludes with a discussion and areas of improvement. Technical details are left to the supplement.

## 2 Materials and methods

Our goal in this work is to create a tool that can predict the toxicity of an arbitrary collection of chemicals for a specific assay. This is in contrast to a model that is fit to a specific mixture using the empirical mixture response. This means our first step is to fit a model to each individual chemical for the assay of interest. We then want to use the set of individual models to predict the mixture effect with uncertainty quantification, and when possible, validate against empirical data. A summary of our approach is shown as pseudocode in Algorithm 1.

**Algorithm 1**: Toxicological Mixture Prediction Algorithm

```
Data: Individual dose response data (c_ik, r_ijk) for chemicals i ∈ 1: N,
      replicate j, and concentration k, Mixture dose C with component
      doses c_1, ..., c_N
Result: (Step 1) Individual dose-response curve parameters and
uncertainty;
      (2) Cluster assignment for
dose-response groups;
(3) Total Mixture dose-response predictions R_mix (Step 1:) Estimate the
individual fixed and random effect parameters for S1 Eq 7 using the
BHM of S2.1 Section in S1 File
```

$$r_{ijk} = \frac{\alpha_i + u_{ij}}{1 + \left(\frac{\theta_i}{c_k}\right)^{\beta_i}} + v_{ij} + \epsilon_{ijk}$$

```
(Step 2:) Compute grouping, if not random: K-means, sill, etc.
(Step 3a): Sample Parameters
for b ← 1 to B do
  Generate a random cluster assignment or use computed cluster: denote
  the grouping as G_b with K_b clusters;
  Sample the remaining parameters α_b = α_{1b:Nb}, etc, from the BHM
posterior;
  Let P_b = {G_b, α_b, β_b, θ_b, u_b, v_b} represent a set of parameters for the b'th
  sampled curve
end
(Step 3b:) Predict Response
foreach P_b with b ← 1 to B do
  Input: Mixture dose C = (c_1, ..., c_N)
  for Cluster k ← 1 to K_b do
    Calculate the group response for the bth sample curve R_{b,k} via RGCA
      using Eq 6 for the inverse f_i^{-1};
```

$$\arg\min_{R_{b,k}} \left| \sum_{q \in Q_k} c_q / f_q^{-1}(R_{b,k}) - 1 \right|$$

```
  end
```

```
Do Calculate total mixture response for the sampled parameters
```
$R_b = \alpha_{max}(1 - \prod_{k_i=1}^{K}(1 - f_{k_i}(C_{k_i})/\alpha_{max}))$, `Eq 3`, using $f_k(C_k) = R_{b,k}$ from the previous step
**end**
**Output:** Median, 5th, and 95th quantiles for the collection $\{R_b\}$ as the estimated mixture response $R_{mix}$ and uncertainty

This section is structured as follows. Section 2.1 reviews the standard methods of Independent Action (IA) and Generalized Concentration Addition (GCA) and how they are combined into a two-step method. A novel extension to GCA is illustrated in Section 2.2, which we call reflected GCA (RGCA). This device is necessary when applying GCA or CA for slope values different than 1 and can also be used in a two-step method. The two-step approach requires parameter estimates and clustering, so the novel extension is followed by a description of the statistical models used for fitting in Section 2.3 and the interpretation of clustering in Section 2.4. Some information is provided about the motivating data for this work in Section 2.5 and we conclude with comments about simulation and validation in Section 2.6.

## 2.1 GCA, IA, and the two-step model

In our notation, $r$ is a response, $c$ a concentration of a chemical, $\alpha$ a maximum response or effect, $\theta$ the concentration to reach 50% of the maximum effect or $EC_{50}$, and $\beta$ the slope. An upper case letter $R$ or $C$ represents the response or concentration of a mixture. Indexing such as $r_i$ represents the $i$'th response out of a collection of $N$ chemicals, while $R_i$ represents the response of a mixture of chemicals with index depending on context. An index $b = 1, \ldots, B$ refers to a sample from the Bayesian posterior where we take $B$ total samples. As summarized in Algorithm 1, $R_b$ refers to the complete mixture response for a particular set of parameters $\mathcal{P}_b$, $R_{b,k}$ refers to the response of the $k$th subcluster for $\mathcal{P}_b$, and $R_{mix}$ refers to the final estimate of the mixture response. In this work we assume a Hill function for a toxic response:

$$r = f(c|\alpha, \theta, \beta) = \frac{\alpha}{1 + \left(\frac{\theta}{x}\right)^{\beta}} \tag{1}$$

The standard version of GCA is derived from concentration addition in [17]. The great advantage of this technique over CA is the ability to make realistic predictions in the case where one chemical has a smaller maximum effect than another. Theoretically, we expect that at high concentrations, the less toxic chemical actually prevents some of the damage from the more effective toxicant. This is the type of antagonism that GCA models, as we will illustrate. The GCA equation for a mixture of two chemicals is written:

$$\frac{[A]}{f_A^{-1}(R)} + \frac{[B]}{f_B^{-1}(R)} = 1 \tag{2}$$

Their method is very flexible in terms of the choice of $f$, the toxic response function, but they mainly work with the Hill model and we build on this specific case.

Recall that a two-step model in our context means that we cluster a set of chemicals and use GCA within clusters as step 1 and assume independent action (IA) across clusters as step 2. Eq 3 expresses IA when the index $k$ corresponds to a single chemical and expresses the two-step model when $k$ corresponds to the $k$'th set of chemicals. Note the introduction of the parameter

$\alpha_{max} = \max_i \alpha_i$ which rescales the final prediction to the highest observed individual response.

$$R = \alpha_{max}(1 - \prod_{k=1}^{K}(1 - \frac{f_k(C_k)}{\alpha_{max}})) \tag{3}$$

Existing theory suggests that chemicals with the same mode of action are expected to obey CA, while chemicals that have unrelated molecular targets are assumed to obey IA [7]. Unfortunately, mode of action is not well understood for many chemicals. Hence we compare the effectiveness of several clustering methods, including no clustering, for our two-step methodology. For completeness we also apply CA, using the same scaling idea that IA applies by resetting all sills to 1, computing the combined effect, and then mutliplying by the largest sill. The slope parameters for CA do not have to be 1 as in GCA.

## 2.2 Reflected GCA

We now make a few observations about the Hill function. First, note that aside for a few special values for $\beta$ like 1 and 1/2, the inverse is not a real number for large values of $r$:

$$c = f^{-1}(r|\alpha, \theta, \beta) = \frac{\theta}{\left(\frac{\alpha}{r} - 1\right)^{1/\beta}} \tag{4}$$

An imaginary component makes the resulting estimate difficult to use and interpret [20]. Next, when $\beta = 1$, we have a formula describing a hyperbola:

$$c = f^{-1}(r|\alpha, \theta, \beta = 1) = \frac{\theta}{\left(\frac{\alpha}{r} - 1\right)} \tag{5}$$

Studying either the plot or the formula, we make the key observation that we can reflect along the axes $x = -\theta$ and $y = \alpha$ to recover the "negative concentration" portion of the graph. This symmetry is a property of hyperbolic functions and illustrated below in Fig 1. The existence of the Hill function for response values above the sill is what makes GCA possible, since this region returns negative concentrations to yield the partial agonist response for large effects.

There is also a secondary symmetry that is crucial for full generality of our proposed approach. Focusing on the continuous portion of the curve that includes positive concentrations (Fig 1, pink box), there is a symmetry between the segment before and after the origin (x = y = 0). This region of symmetry is bounded by the reflection axes and X-Y axes as illustrated in Fig 1 with the pink and blue boxes. This additional symmetry is relevant for two reasons. First, the negative reflected component (teal line) of the defined curve (green line) described earlier does not extend across the full negative support (purple line). By reflection, the maximum defined effect is only two times the sill, so if some other chemical in the mixture has a maximum effect above this threshold, there is no defined value. Hence, we need to define the extension (red line) and reflect it as well (purple line). The second reason the extension is necessary is because the sill parameter in the observed dose-response can be negative. In this case, the relevant portion for inversion at positive effects is the extension shown as the red line, since the entire curve is flipped across the X-axis.

Our first novel contribution is to apply this reflection argument when the slope parameter $\beta$ is not 1. The inverse function is specified for this domain differently from the regular inverse and the derivation is provided in the next section, resulting in Eq 6 when $\alpha > 0$. Similar

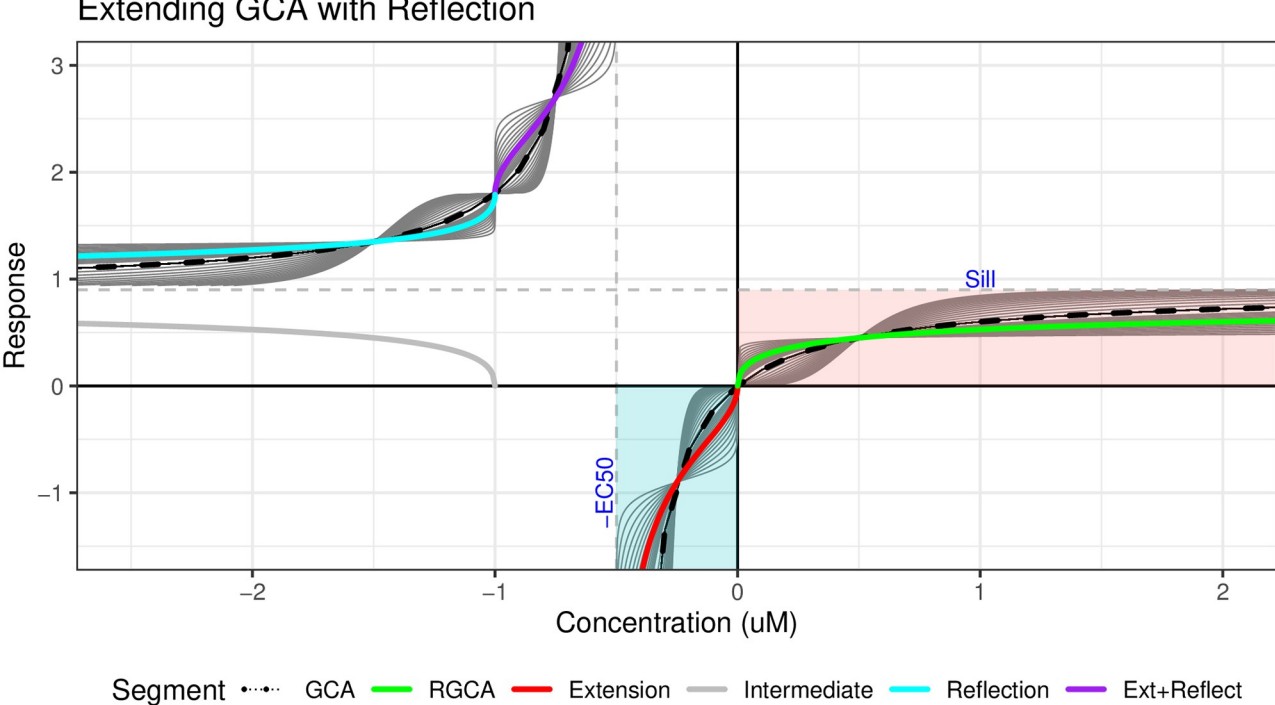

**Fig 1. Hill function symmetry.** Reflections can be used to extend the Hill function with a slope $\neq 1$ in green to match the support of the standard GCA case, shown as a dashed black line. The green line (RGCA) has parameters (Sill, $EC_{50}$, Slope) = (0.9, 0.5, 0.5 while the shadow lines have slope values ranging from 0.1 to 4. The teal line is acquired by reflecting the green segment across the vertical line Y = -$EC_{50}$(Intermediate step, gray) and then across the horizontal line X = Sill. The Extension (red line) is found by flipping the pink box containing the RGCA line across the diagonal Y = -X (not shown) and using the $EC_{50}$ value as the effective sill. The Reflected Extension (purple line) is found by the same procedure as the Reflection (teal) line, starting with the Extension (red) line.

equations hold for the case $\alpha < 0$ with appropriate sign changes for each region, see the supplement.

This inverse provides a wide enough support to satisfy the invertibility requirements of Eq 2, so we now have the ability to apply GCA to Hill models with non-unit slopes.

**2.2.1 Derivation of RGCA.** Here, we derive how to extend GCA to non-unit slopes. To ensure RGCA is a valid function that is fully-defined across the real-number domain, we utilize geometric techniques such as reflections, extensions, and substitutions. Starting with the three parameter Hill model in Eq 1 and its inverse, we define three additional functions as combinations of reflections and extensions.

*Reflection*. The first function is a reflection (teal line, Fig 1) along the axes defined by the parameters $x = -\theta$ and $y = \alpha$. The calculations are straightforward since the axes are parallel to the standard X and Y axes. We first subtract the offset for each axis, then negate the sign for the corresponding variable, and then add back the offset. We represent the concentration as $c$ and the reflected concentration as $c'$, and similarly for the response $r$ and $r'$:

$$c' = (-\theta) - (c - (-\theta)) = -2\theta - x$$

$$r' = \alpha - (r - \alpha) = 2\alpha - r$$

To provide the inverse, we just plug in the new $(c', r')$ into the relation $f^{-1}(r) = c$:

$$-2\theta - c = f^{-1}(2\alpha - r) \Rightarrow c = -2\theta - \theta\left(\frac{\alpha}{2\alpha - r} - 1\right)^{-1/\beta}$$

*Extension.* Next we extend (red line, Fig 1) the Hill function to negative values of $c$ between the origin and the asymptote $c = -\theta$. The key observation is that, while it is not a direct reflection along an axis like y = x, it can be thought of as a separate Hill function where the role of the sill and EC$_{50}$ have been reversed along with $c$ and $r$. Hence, the standard Hill function becomes the inverse, where the two parameters are swapped and the new sill (previously the EC$_{50}$) and input response are negated to account for the sign change of the support. While it is necessary to use the EC$_{50}$ as the sill (so that there is a vertical asymptote at $c = -\theta$), it is not necessary to use the sill as the new EC$_{50}$.

$$c = f_{ext}^{-1}(r) = \frac{-\theta}{1 + \left(\frac{\alpha}{-r}\right)^{\beta}}, \qquad r = f_{ext}(c) = -\alpha\left(\frac{-\theta}{c} - 1\right)^{-1/\beta}$$

Here the equation for $x$ as a function of response $R$ has the form of a Hill function with slope $\beta$ even though it represents the inverse; the equation for the response $R$ as a function of $c$ has the form of the Hill inverse. Finally we create a smooth transition by inverting the slope parameter to match the slope of the adjacent segment:

$$c = f_{ext}^{-1}(r) = \frac{-\theta}{1 + \left(\frac{\alpha}{-r}\right)^{1/\beta}}$$

If the slope is not inverted, the resulting curve will have a non-smooth kink at 0 and lead to situations where there is no solution to the RGCA equation.

*Reflected Extension.* Finally, we reflect the extension we just derived over the axes $x = -\theta$ and $y = \alpha$ (purple line, Fig 1). The method is the same as before so we plug $c'$ and $r'$ into $f_{ext}^{-1}(r) = c$

$$-2\theta - c = f_{ext}^{-1}(2\alpha - r) \Rightarrow c = -2\theta + \frac{\theta}{1 + \left(\frac{\alpha}{r - 2\alpha}\right)^{1/\beta}}$$

*Combined Inverse.* Putting the pieces together, we have a piecewise inverse for the four domains:

$$c = f^{-1}(r|\alpha > 0, \theta, \beta > 0) = \begin{cases} \dfrac{-\theta}{1 + (\frac{-\alpha}{r})^{1/\beta}} & r \in (-\infty, 0) \\[3mm] \theta(\frac{\alpha}{r} - 1)^{-1/\beta} & r \in [0, \alpha) \\[3mm] -2\theta - \theta\left(\dfrac{\alpha}{2\alpha - r} - 1\right)^{-1/\beta} & r \in (\alpha, 2\alpha) \\[3mm] -2\theta + \dfrac{\theta}{1 + (\frac{\alpha}{r - 2\alpha})^{1/\beta}} & r \in (2\alpha, \infty) \end{cases} \qquad (6)$$

The resulting inverse maintains a coarse hyperbolic shape and continuity and is smooth at the

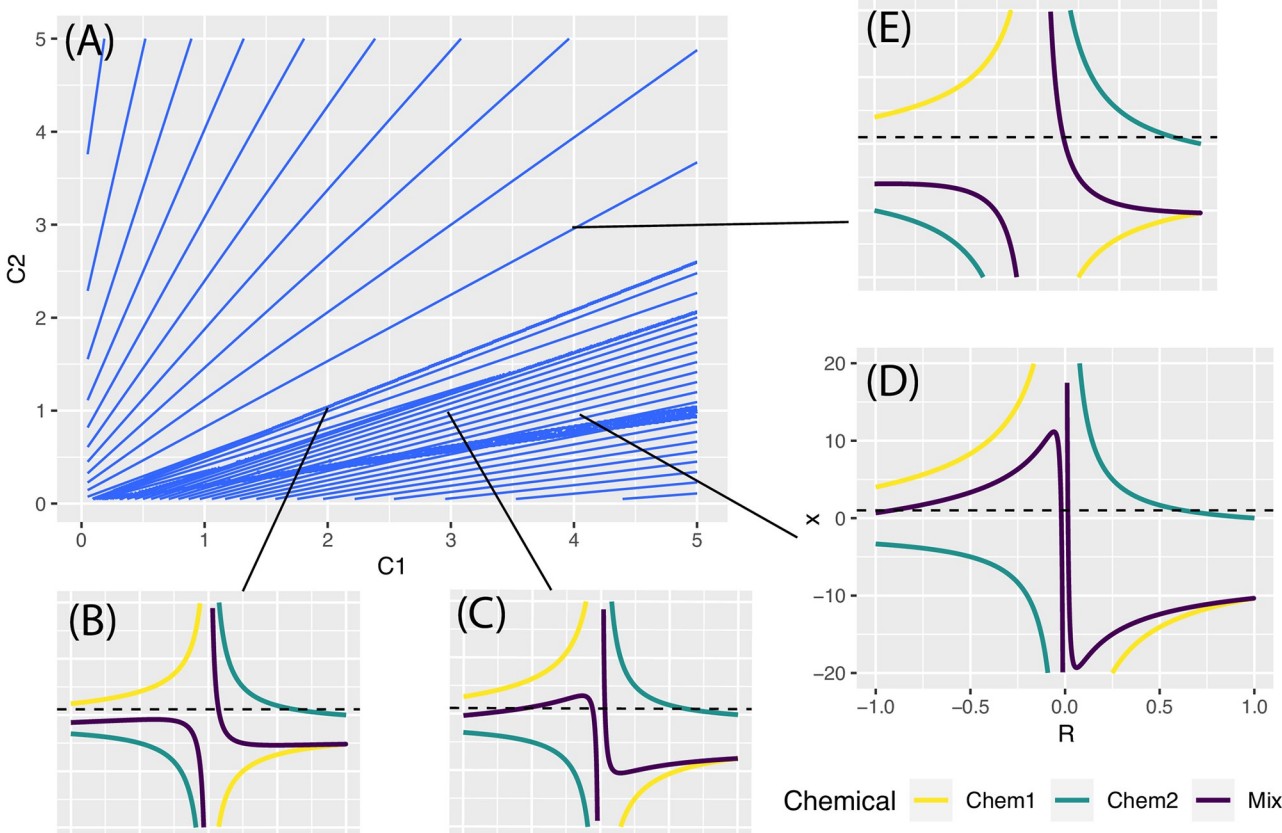

**Fig 2. Multiple solutions with RGCA.** Isobologram (A) with inset plots showing curves used to solve RGCA based on Eq 2. The isobologram shows lines of equal mixture effect for different concentrations of two chemicals. In each inset plot, the line for chemical 1 represents the term $[A]/f_A^{-1}(R)$ and similarly for chemical 2, while the line for Mix represents the sum of chemicals 1 and 2 under the assumption that the concentrations $[A]$ and $[B]$ correspond to the point shown in the isobologram. A solution for the mixture corresponds to the Mix line intersecting the horizontal dotted line at x = 1, representing the right side of Eq 2. The solution in the subplots equals the value for the contour line in the isobologram. Chemical 1 in yellow has parameters (Sill, $EC_{50}$, slope) = (1, -2, 1.5) while chemical 2 in green has parameters (0.6, 1, 1). Plot (E) with concentrations (c1, c2) = (4, 3) is comparable to GCA, while the lower plots show a progression from one to three solutions and a corresponding phase change in the isobologram caused by the bifurcation. With enough of chemical 1, the predicted mixture response is either near zero or negative.

transitions. This procedure is not limited to the Hill function and can be applied to any monotonic dose response function, but the resulting stability may vary. Note that negative slope parameters for the Hill function are not supported. The negative sill case $\alpha < 0$ is shown in the appendix for completeness.

**2.2.2 Limitations.** An important property of GCA that partially carries over to RGCA is the existence and uniqueness of a solution. This means that a set of chemicals with arbitrary sill and EC50 values will have a single solution for the equivalent response $r$ in Eq 2. This stability is due to the fact that slope values are fixed to 1, and that it is extremely unlikely that the sills and $EC_{50}$ values will perfectly balance out. In the case of RGCA, it is possible that the solution may not be unique if any chemical's sill is negative.

A proof sketch for the existence and uniqueness of the GCA solution is provided in the supplementary material. RGCA has a provable and unique solution when all of the chemicals have positive sill value, which is applicable across a wide variety of problems in mixtures toxicology. The problem for RGCA arises when the slope values are not all 1 and some sill values are negative. In this case there can be a "battle" where one chemical dominates in one region and

another chemical dominates in another. This complex behavior is shown in Fig 2, which shows an isobole plot (A) for two chemicals with a clear change in appearance near the bottom half. Chemical one (C1) has (sill, slope, $EC_{50}$) of (-2, 1.5, 1) and chemical two (C2) has parameters (1, 1, 0.6). The four smaller inset plots, B to E, show RGCA coefficients $1/f_i^{-1}(R)$ from Eq 2 for specific concentrations pointed out by the black lines. The dotted horizontal lines in each inset plot represents 1, the intersection of which provides the solution and matches the value of the contour in the isobole plot. Plot (E) in the top right for concentrations (c1 = 4, c2 = 3) are similar to the result of GCA, which always has one solution. The bottom plot progression (B,C,D) shows how the RGCA solution changes from one unique value to three unique values as the proportion of chemical 1 to the total mixture increases. With chemical 2 at a fixed concentration, chemical 1 becomes the primary driver of the mixture and introduces a negative response. The boundary between regions is a bifurcation point where there is just enough chemical 1 to create a second solution (but not a third). Mathematically, the additional solutions are just the result of the slope parameters determining the rate at which particular functions go to zero or infinity.

Numerical precision can be an issue. For example, in the bottom right inset plot for (c1 = 4, c2 = 2), a local optima at the positive boundary or a negative optima may be found rather than the true positive optima near the origin. This is the reason for the denser region in the lower half of Fig 2. Similar patterns can be seen in S1 Fig in S1 File in the supplement. Extending to larger mixtures is expected to yield similar results, as all positive sill terms can be added together and likewise for negative terms, reducing to the binary case shown.

In summary, when the sills of the chemicals are different signs, some combinations of parameters lead to multiple solutions. Fig 2 suggests a linear boundary beyond which multiple solutions and numerical issues may be a problem, but a general formula for when there are three solutions does not appear tractable. To avoid ambiguity, we could choose to always select the positive solution if available, but it is not clear that a positive solution near 0 is more correct than a negative solution with larger magnitude. Since the numerical issues are difficult to predict, our recommendation is to simply note any optimization messages or errors when using this method.

## 2.3 Bayesian data model

As illustrated in Fig 3, chemicals have replicates and high variability between replicates. We express the Hill function as a statistical model by introducing random effect and noise terms $(u, v, \epsilon)$ and use additional indices to track the chemical ($i$) and replicate ($j$) and concentration sample ($k$):

$$R_{ijk} = \frac{\alpha_i + u_{ij}}{1 + \left(\dfrac{\theta_i}{x_k}\right)^{\beta_j}} + v_{ij} + \epsilon_{ijk} \tag{7}$$

The noise term $\epsilon$ represents the standard assumption of independent and identically distributed Gaussian noise and allows for negative values of the response $R$, which are observed in the data even though we might presume that negative effects are unrealistic. The term $u_{ij}$ is a random effect that accounts for the replicates having variable maximum effects. The $v_{ij}$ term acts as a y-intercept random effect; naturally when the concentration is 0 we expect a response of 0, but due to the data collection procedure this is not guaranteed.

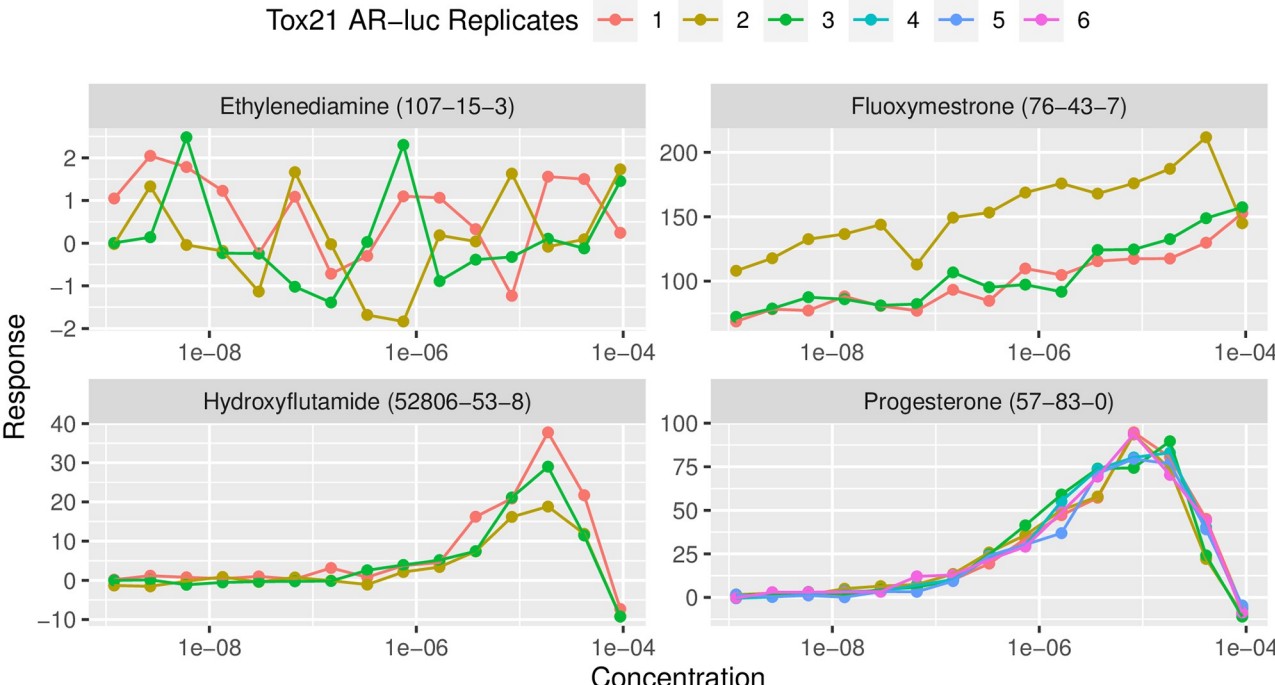

**Fig 3. Examples of Tox21 data.** Example dose response curves from the Tox21 data set with chemical name (CAS number). Clockwise from the top left, each plot shows replicates exhibiting: no effect, a random intercept effect, a random sill effect, and a consistent response without random effects. Eq 7 can account for all of the random effects, but we do not explicitly model a no-effect response with a null model because we include all chemicals when predicting a mixture response, even if there is no effect. A sharp decrease in response implies cytotoxicity and such points are removed before analysis.

Since this model does not have a closed form solution for the parameter estimates, and we care about the uncertainty in the estimates, we fit a Bayesian model using Markov Chain Monte Carlo, see [21] for a full reference. Note that the variance of the noise term is specified for each chemical rather than globally. This was done for two reasons: there are at least three replicates for each chemical, and a global variance parameter can be influenced by a chemical with a large sill, while chemicals with small but noticeable effects will get treated as noise. Using 25,000 iterations, posterior estimates of the parameters are found using a burn-in of 5000 and thinning the remaining 20,000 iterations by taking every 20th sample and then taking the median as a robust estimate of the center of the distribution. For sampling curves, we sample directly from the thinned set of posterior samples as described below in section 2.1. The priors for the main parameters are Gaussian with mean 0 and variance 200 for the sill, Gamma with mean and variance 1 for the slope, and Gamma with mean 1 and variance 100 for the $EC_{50}$, respectively. The sill and $EC_{50}$ priors are weakly informative to avoid absurdly large values when the dose response curve is incomplete [22]. The slope parameters are given Gamma priors centered at 1 because our method requires a positive slope and a center of 1 reflects the baseline assumption from GCA. The full model specification is in the Supplement, S2 Section in S1 File, and R code is available on Github.

An immediate concern may be that our model does not allow for a null or no-effect chemical. This is a deliberate choice. In the context of CA, the natural thing to do is to assume extreme parameters, such as a very small sill, a very large $EC_{50}$, and fix the slope to 1, or simply ignore the no-effect chemicals. We make an assumption that all chemicals are relevant so we do not want to throw any out. We also intend to use the slope parameter for clustering, so the

choice of a null model becomes consequential. Rather than attempt to justify a fixed null model, we leave it to the algorithm to find the best fit. We comment in the conclusion that this question is a direction for future work.

## 2.4 Clustering as interpolation

Clustering is a nuanced task that involves many decisions. Functional data such as dose response curves can be clustered according to the parameters or prior information, with multiple settings to adjust depending on what specific cluster method is used. In the context of the mixture problem, the benefit of clustering for a two-step or joint method is to serve as an interpolation between IA and CA [13]. A simple but motivating example is a mixture composed of androgen receptor (AR) agonists and estrogen receptor (ER) agonists. If this mixture is applied to an an AR assay, one would expect the significant effect to come from the AR agonists. However, a naive application of CA on this kind of mixture will significantly underestimate the predicted response because the ER agonists have low or zero responses individually that dilute the effect.

In some cases like the example given, there is a clear candidate for clustering method. But as we will show in the Results, in practice this clustering isn't enough to correctly recover the mixture response because the independence assumption ignores molecular dynamics such as binding affinity or synergy, and is also unaware of whatever grouping would result from knowing the true modes of action. As a secondary contribution of this work, we explore the effectiveness of a few different clustering methods to improve predictions when true modes of action are not known.

The first clustering approach we consider is to group chemicals on the binary condition of having a positive response versus a non-positive response. This is a simple clustering on the sill parameter. A second approach we consider is a partial random clustering. The chemicals are randomly assigned into one of two to five groups (hence partial, rather than allowing any possible group), along with one assignment representing the single group of CA and one assignment to represent the completely separated grouping of IA. The last cluster approach we explore is a K-means grouping based on all of the parameters, a logical extension of the argument that zero-effect chemicals should be grouped separately from positive-effect chemicals. The assignments are computed using the posterior median parameter estimates and a few reasonable options are taken based on the Elbow criteria (the cluster count beyond which additional clusters do not significantly decrease the within-group variability).

Our two-stage approach can now be summarized as follows (See Algorithm 1). We first fit Hill functions with random effects to the individual chemicals using a Bayesian model. To make a single prediction, we sample the parameters from the posterior distributions for each chemical, propagating variance by adding noise according to the estimated random effect. If the clustering method has any randomness (eg random clustering), we sample a cluster assignment. Given the parameters and a clustering assignment, we apply RGCA to the chemicals in each group to get a combined group effect, and then apply the IA model of Eq 3 to the set of group effects. GCA and CA predictions rely only on posterior mean parameter estimates for purposes of comparison, but could be adapted to use the Bayesian approach.

The uncertainty quantification for a predicted effect at a given concentration is achieved by generating multiple predictions (eg $B = 100$)) and estimating point-wise quantiles. For comparison to existing methods, we compute curves for regular IA where every chemical gets its own cluster and GCA and CA where all chemicals are in one cluster. For GCA we fix the slope values to 1 while CA scales all sills to 1 and applies GCA, multiplying the final prediction by the largest sill. IA uses all of the same fitted parameters as our two-step method. For these

standard methods, we do not compute uncertainty intervals to improve contrast with our Bayesian method. In practice, all of these techniques can be used with the Bayesian framework and will result in similar uncertainty intervals.

## 2.5 Tox21 data

Tox21 [23] is a library of 10,000 chemicals selected for their potential relevance to health and amenability to testing. Quantitative high throughput screening (qHTS) with specific assays is used to efficiently find which chemicals are relevant for pathways or targets of interest [19].

We focus on a subset of Tox21 that evaluates responses with the androgen receptor luciferase (AR-luc) assay. AR-luc is a cell line derived from human breast cancer cells that includes a gene to express a firefly luciferase protein. A promoter, MMTV, is introduced into the cell line DNA as the response element for the AR, while the luciferase gene is added as a reporter in the regulatory sequence with MMTV. In other words, MMTV is used as the piece of DNA (promoter) that the AR first attaches to in the nucleus (regulatory sequence) to begin transcription, and the luciferase gene is the subsequent section that is transcribed to create a measurable effect (reporter). This corresponds to the last step of the AR pathway, right before RNA transcription begins, so the data collected should be highly correlated with a count of how often a gene is expressed or protein produced. See [24, 25] for additional details on the analysis of Tox21 data for the AR-luc assay.

The input data consists of individual dose responses for 18 chemicals. Six of the chemicals are clear AR agonists, one is a weak agonist, and the remaining 11 are known estrogen receptor (ER) agonists but appear to have no effect on AR. Other assays in Tox21 suggest that some of these known ER agonists can act as AR antagonists. There are dose responses for 69 different mixtures as part of the same Tox21 data set, including binary, equipotent, equiconcentration, and combinations dominated by one or two chemicals. Some of the mixtures include only the AR or ER agonists. Many mixtures were designed by referring to their activity in a previous study, with chemical concentrations in the mixtures at different proportions of their $EC_{50}$ values [25]. As a part of quality control of the original data, doses that exhibited cytotoxic responses were flagged. These data points were subsequently excluded in our analysis.

Because our approach does not account for synergy or direct antagonism, the mixtures that are more sensitive to these interactions are expected to have greater prediction error. In fact, some binary mixtures were found to have antagonistic responses that entirely ignore the effect of one toxic compound. Furthermore, clustering for a binary mixture reduces to either GCA (i.e. one cluster) or IA (i.e. two clusters). Our focus is on "large" mixtures that contain proportions of many chemicals, which corresponds to a more realistic exposure scenario in which humans are exposed to a wide variety of complex chemicals and stressors through their life course—i.e. a complete exposome [26]. Additionally, with large, complex mixtures, the effects of synergy or antagonism are less likely to impact the whole mixture response [27]).

A positive control chemical is used as a benchmark to define a response of 100 while a vehicle control defines a response of 0. The response does not have defined units because the response is a measure of relative fluorescence, and negative values simply mean less response than the vehicle control.

## 2.6 Simulation and validation

We conduct a simulation study to explore relative performance of RGCA and/or a two step approach. A single simulation run consists of generating Hill parameters and assigning a cluster grouping for 10 or 20 chemicals, treating this as the true mixture model. The models are then fit using the true parameters where possible and the clustering according to the method.

**Table 1. Simulation setup.** Parameters for the mixture response are sampled uniformly from the specified intervals according to the true model.

| True Model | Slope Range | Sill Range | Clustering |
|---|---|---|---|
| GCA | 1 | [1.5, 10] | None |
| RGCA (slope $\sim 1$) | U[0.5, 1.5] | U[1.5, 10] | None |
| RGCA (slope $>> 1$) | U[0.1, 10] | U[1.5, 10] | None |
| RGCA (sill $< 0$) | U[0.5, 1.5] | U[-10, 10] | Postive vs non-positive sills |
| RGCA (2-step) | U[0.5, 1.5] | U[1.5, 10] | Random, 2–3 clusters |
| RGCA (2-step KM) | U[0.5, 1.5] | U[1.5, 10] | K-Means on all parameters |

Mixture doses are computed as equipotent vectors for a range of dilutions. For example, a GCA prediction will use the correct parameters but fix the slope values to 1 and assume a single group (all chemicals in one cluster). There are 50 simulation runs per model per mixture size. The set of true models is describe in Table 1. The $EC_{50}$ parameters are sampled uniformly from 0.1 to 20 for all models. The models used for fitting include the true models along with CA and IA. When the fitted model is the true model, we expect 0 error, with the exception of the random clustering method which always creates a random grouping. When the random model is used as the truth, only one random grouping is chosen to generate a mixture response; when used as fitting procedure, ten random groupings are pre-generated with four assignments having two clusters, four assignments having three clusters, and an assignment each for CA and IA. When GCA is the truth, RGCA should yield identical results because RGCA generalizes GCA. Under RGCA, we expect a large error for all other methods when the slope parameter is different from 1. It is not clear how much relative error to expect when the two-step model is the true model or used for fitting.

We do not perform a simulation study for the mixed effect model, as this performance can be evaluated empirically and is compared to a simpler approach of maximum likelihood using an existing dose response curve fitting package called `drc` from [28].

For the data application, we evaluate the same methods we simulated but expand the random clustering from ten to 100 possible assignments to match the number of samples $B = 100$ used to generate uncertainty intervals: 1 each for GCA and IA, then 10 that group among 2 clusters, 20 that group among 3 clusters, 30 that group among 4 clusters, and 38 that group among 5 clusters. The number of assignments per cluster count is arbitrary and roughly reflects the fact that there are many more possible assignments when the number of clusters increases.

Error can be quantified with mean squared error (MSE), log likelihood (LLH) or a continuous ranked probability score (CRPS). CRPS is a proper scoring rule: the true model will give the best score (but other models may give equally good scores) [29]. Intuitively, a CRPS measures how well the current distribution predicts the observed value in comparison to a perfect predictive distribution which puts all mass on the observed value. The empirical CRPS at a concentration $x_j$ with observed response $R$ is computed as

$$\text{CRPS}(F_j, R) = \sum_{i=1}^{N}[\hat{F}_j(y_i) - 1(y_i > R)]^2, \qquad \hat{F}_j(y) = \frac{1}{N}\sum_{i=1}^{N}1(f_i(x_j) < y) \qquad (8)$$

This equation describes the empirical cumulative distribution function $F_j$ of the predicted responses $y_i$ at point $x_j$, for index $i$ representing the sampled curves.

When the prediction is far from the truth, the CRPS score may give identical results for all methods, so we also include LLH and MSE. LLH is computed using a kernel density estimate

rather than the empirical CDF used in the CRPS:

$$\text{LL}(R) = -\log \frac{1}{N} \sum_i \phi(R | y_i, \sigma_{bw}) \tag{9}$$

The kernel $\phi$ is a Gaussian centered at the value $y_i$ with a computed bin width. When there is only one data point, the LLH is not provided since it depends entirely on the bin width. In cases where the data is far from the truth, the LLH may be infinity (representing a likelihood that is effectively 0). MSE for our data is computed as

$$\text{MSE} = \sum_{i,j} (R_{ij} - f(x_{ijk}))^2 \tag{10}$$

For the simulation study, since we are only testing the applicability of various models to each other rather than the statistical accuracy of a fitting procedure with noisy data, so we compare fits using MSE. All three scores are computed for our predictions with the Tox21 data, which is used for validation. The R package `scoringRules` described in [30] is used to compute the empirical CRPS and LLH. Each score emphasizes a different quality. MSE ignores uncertainty and just tells how closely we recovered the observations; CRPS emphasizes both centering and uncertainty; LLH is more lenient when it comes to the mean or center being wrong as long as the uncertainty is wide enough to represent the data.

## 3 Results

### 3.1 Simulation study

Our simulation study illustrated in Fig 4 shows that RGCA and two-step methods can be helpful for improving predictions when the true model is not GCA. We can make a number of qualitative comparisons because of the variety of true models we tested. For example, the error between GCA (as truth) and IA is roughly twice the error between GCA and a joint model, which is reasonable since the joint models interpolate between GCA and IA. When slope parameters are near 1, GCA and RGCA are nearly identical and the two-step methods have large errors; when slope parameters are allowed to vary from 0.1 to 10 (slope $>>$1), the error of GCA and two-step method become comparable. These observations suggest that extreme slopes have an interpolating effect similar to the two-step methods, decreasing the difference between GCA and IA. A similar conclusion seems to hold for simulations that allow negative sills, an edge case discussed in Section 2.2.2. When the two-step methods are the true responses, all of the methods have less error, again due to the interpolation effect. It is surprising that our implementation of CA has the lowest error among one-step approaches when the true model is two-step, especially in larger mixture sets.

The main conclusions from this simulation study are that 1) RGCA alone may not provide much benefit if the slopes are around 1 and sills have the same sign, and 2) a two-step approach using RGCA with random clusters tends to perform well in a variety of settings, including unknown groupings into modes of action. The benefits often increased with the size of the mixture, but our study looked at equipotent dose vectors. If a mixture is driven by a few potent chemicals, the effective mixture size and resulting errors may be smaller.

### 3.2 Parameter estimates and clustering

The parameter estimates for our Bayesian random effect model and for the maximum likelihood found with the R package `drc` is shown below in Table 2. For the chemicals with large responses, the models arrive at nearly identical results. The random effect model has different

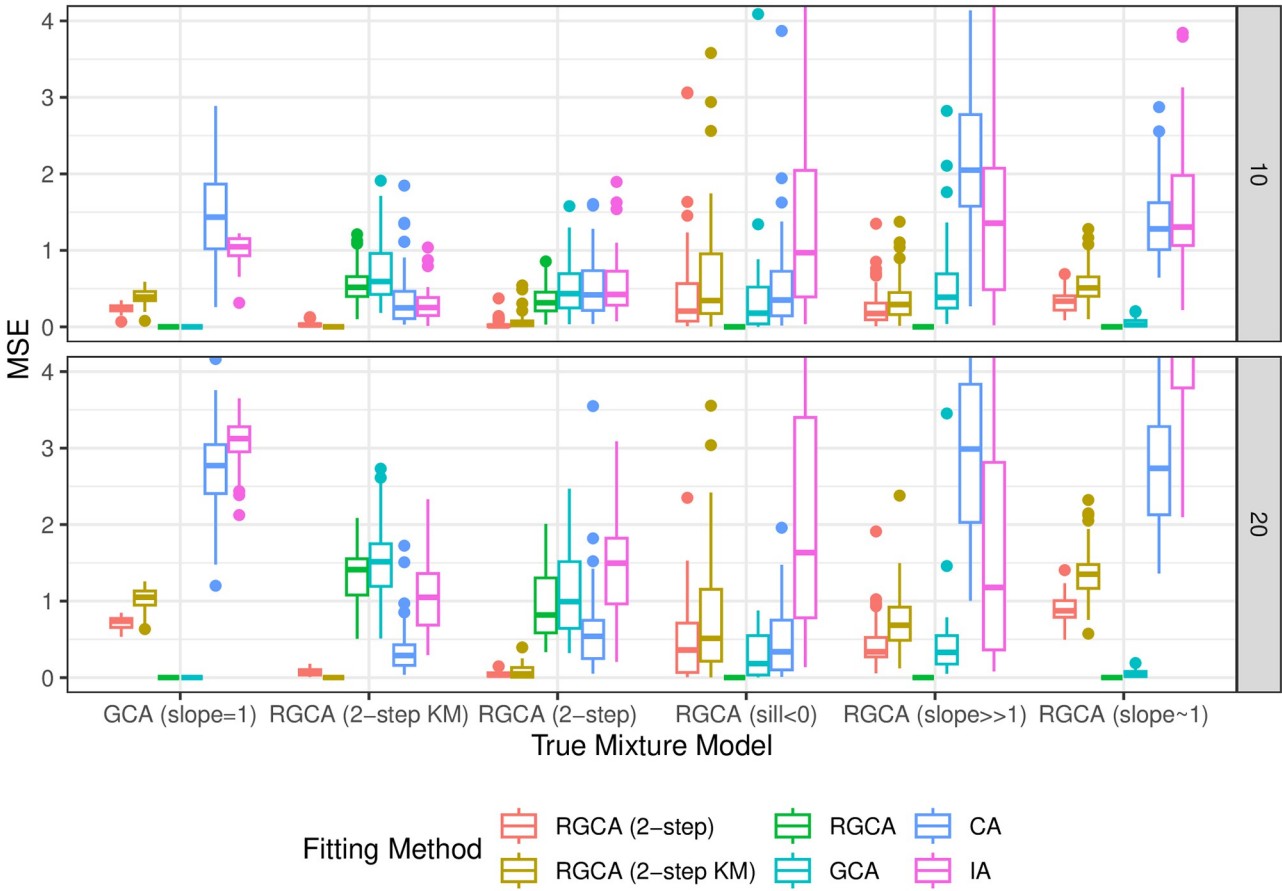

**Fig 4. Simulation study results.** A simulation study suggests that a two-step approach with cluster uncertainty cannot be adequately substituted with either GCA or IA. Mean Square Error (MSE) and Continuous Rank Probability Score (CRPS) are used to compare predictions of simulated mixtures and lower scores are better. The scores for the simpler methods deteriorate as the size of the mixture increases, suggesting that the benefit of our method increases with larger mixtures. Mixture sizes are n = 10 and n = 20 as shown on the right side of each plot and each box consists of 50 simulations.

sill estimates compared to `drc` because all of the curve effects are averaged for `drc`, while our model takes just one replicate and adds offsets (realizations of random effects) for the other replicates. The differences between the small or no effect chemical parameters is due to the regularizing effect of priors which were chosen to prevent extreme values and ambiguities in the likelihood. For example, note that the 17th chemical (benzyl butyl phthalate) has an estimated sill of 1.175 and an $EC_{50}$ of 2e-12 under our method, suggesting a no-effect chemical, while the `drc` package finds a sill of -7500 and an $EC_{50}$ of about 4e-3. Both parameter sets explain the data and offer similar likelihood, but extrapolated responses for higher doses could be more misleading with the `drc` parameters. In the supplement, see S2 Table in S1 File for additional parameter estimates, S2 Fig in S1 File for trace plots showing convergence of parameter samples to stationary distributions, and S3 Fig in S1 File for a visualization of the resulting curve fit.

Because of identifiability or flat likelihood issues, some of the MCMC chains do not exhibit convergence to a stationary distribution. In brief, for no-effect curves, one could set the sill to 0 and the other parameters could be completely free. While this could pose an issue in our sampling by creating a sample curve with a non-zero sill and an extreme $EC_{50}$, we find that

**Table 2. Fitted parameters for Tox21.** Comparison of parameter estimates from the random effect model and the maximum likelihood of `drc`.

| CAS | Name | Sill | $EC_{50}$ | Slope | Sill (drc) | $EC_{50}$(drc) | Slope (drc) |
|---|---|---|---|---|---|---|---|
| 107–15–3 | Ethylenediamine | 8.91 | $2.55e-01$ | 1.12 | 0.60 | $2.35e-06$ | −0.05 |
| 143–50–0 | Kepone | −18.10 | $1.16e-04$ | 1.60 | −10.07 | $4.99e-05$ | 3.46 |
| 15972–60–8 | Alachlor | −38.42 | $4.98e-04$ | 0.81 | −58.53 | $7.87e-04$ | 0.77 |
| 17924–92–4 | Zearalenone | −30.38 | $3.26e-04$ | 0.70 | −25.75 | $2.21e-04$ | 0.68 |
| 34256–82–1 | Acetochlor | −27.35 | $2.83e-04$ | 0.85 | 1.02 | $2.24e-06$ | −4.51 |
| 434–07–1 | Oxymetholone | 90.34 | $6.20e-09$ | 1.16 | 93.00 | $5.91e-09$ | 1.14 |
| 50–02–2 | Dexamethasone | 357.85 | $5.52e-09$ | 1.05 | 398.32 | $5.12e-09$ | 0.90 |
| 50–29–3 | p,p'-DDT | −25.05 | $1.07e-04$ | 4.94 | −64.83 | $1.63e-04$ | 3.40 |
| 52806–53–8 | Hydroxyflutamide | 33.58 | $4.01e-06$ | 1.83 | 21.85 | $3.49e-06$ | 2.22 |
| 57–83–0 | Progesterone | 96.79 | $1.45e-06$ | 0.88 | 98.23 | $1.47e-06$ | 0.87 |
| 63–05–8 | 4-Androstene-3,17-dione | 109.96 | $3.19e-08$ | 0.89 | 110.67 | $3.25e-08$ | 0.88 |
| 71–58–9 | Medroxyprogesterone acetate | 238.26 | $7.87e-09$ | 0.59 | 265.92 | $6.76e-09$ | 0.53 |
| 76–43–7 | Fluoxymestrone | 152.85 | $7.74e-09$ | 0.17 | 274.83 | $5.14e-06$ | 0.10 |
| 80–05–7 | Bisphenol A | −17.06 | $5.67e-04$ | 0.55 | −4.95 | $1.31e-05$ | 0.89 |
| 80–43–3 | Dicumyl peroxide | −10.72 | $3.17e-03$ | 0.51 | 0.86 | $1.77e-07$ | −1.89 |
| 84852–15–3 | 4-Nonylphenol, branched | −31.41 | $2.05e-03$ | 0.52 | −30.07 | $7.15e-04$ | 0.59 |
| 85–68–7 | Benzyl butyl phthalate | 1.17 | $2.08e-12$ | 0.78 | −7502.20 | $4.24e-03$ | 9.24 |
| 90–05–1 | 2-Methoxyphenol | 20.11 | $8.27e-02$ | 0.34 | 2.27 | $3.12e-05$ | 0.43 |

practically it doesn't cause an issue because the chance of a bad combination of parameters is low. Very strong priors could be used to improve identifiability, but at a cost of introducing additional bias. See S1 Table and S2 Fig in the S1 File for an analysis of the MCMC convergence to stationarity.

### 3.3 Application results

For the 17 Tox21 mixtures tested, a joint or two-step method with RGCA was superior to GCA and CA for a majority of the cases. In particular, our predictions with RGCA were better for most of the mixtures tested using the CRPS metric and about half of the mixtures based on MSE, see S3 and S4 Tables in S1 File respectively. LLH is included for completeness in S5 Table in S1 File. When comparing just RGCA with random clustering to GCA, the RGCA joint method was superior in 14 out of the 17 mixtures by both CRPS and MSE. The best results relative to GCA and CA were with mixtures which include large concentrations of the ER agonist (4x $EC_{50}$) in addition to the AR agonists. We had less success with smaller mixtures, such as when the ER agonists were excluded, suggesting that mixtures with more chemicals may help cancel out synergistic or antagonistic effects. See S6 Table in S1 File for descriptions of the mixtures. We illustrate the scores in aggregate as boxplots in Fig 5.

Although our methods are significantly better than GCA or CA, IA proved surprisingly effective at predicting the mixture response, especially when measured by MSE. When measured by CRPS, the results are comparable to our joint methods and occasionally inferior. This illustrates a trade off where a method with some error but with uncertainty quantification can do a better job of explaining all of the data compared to a method that predicts the center of the data well but does help understand the spread. A series of representative figures, Figs 6–8, is shown below. See the supplement, S4 and S5 Figs in S1 File for additional plots across the tested mixtures.

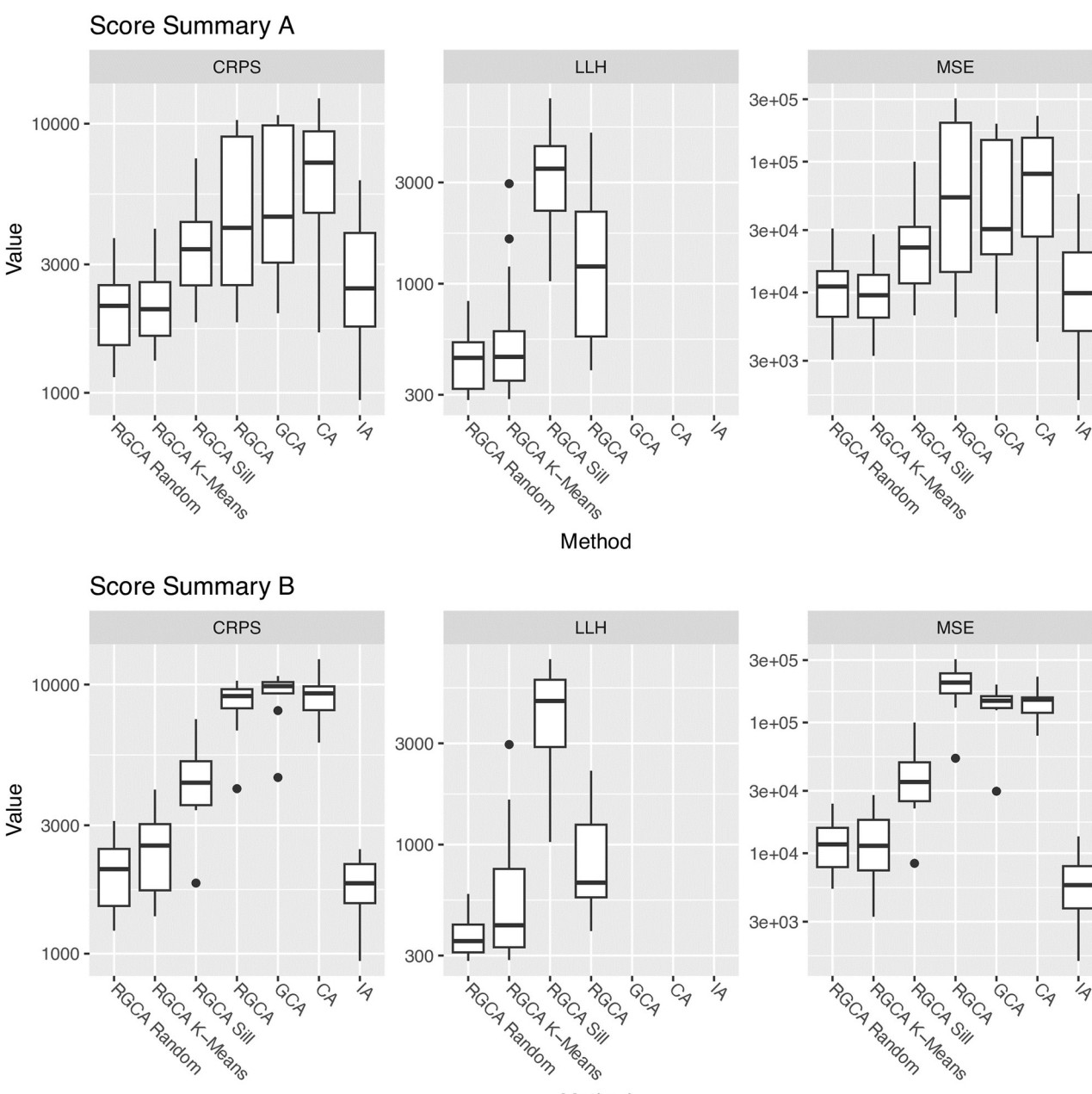

**Fig 5. Application results.** Comparison of methods applied to the Tox21 dataset, lower scores are better. We compare a total of six methods: CA, IA, GCA, RGCA, RGCA grouped by sill, clustering by K-means on all parameters, and random clustering. The scoring metrics are the continuous rank probability score (CRPS), mean squared error from the median curve (MSE), and log likelihood (LLH). Score Summary A at top shows scores across all of the mixtures in tables S3-S5 Tables in S1 File from the supplement. Summary B is restricted to mixtures with large doses of ER agonists: mix 12, 20, 43, 52, 55, 57, 62. Note log-scaled y-axis. GCA, IA, and CA do not have LLH values since they were computed without uncertainty. IA and the two-step methods with RGCA perform best, suggesting that larger mixtures are following IA more than CA.

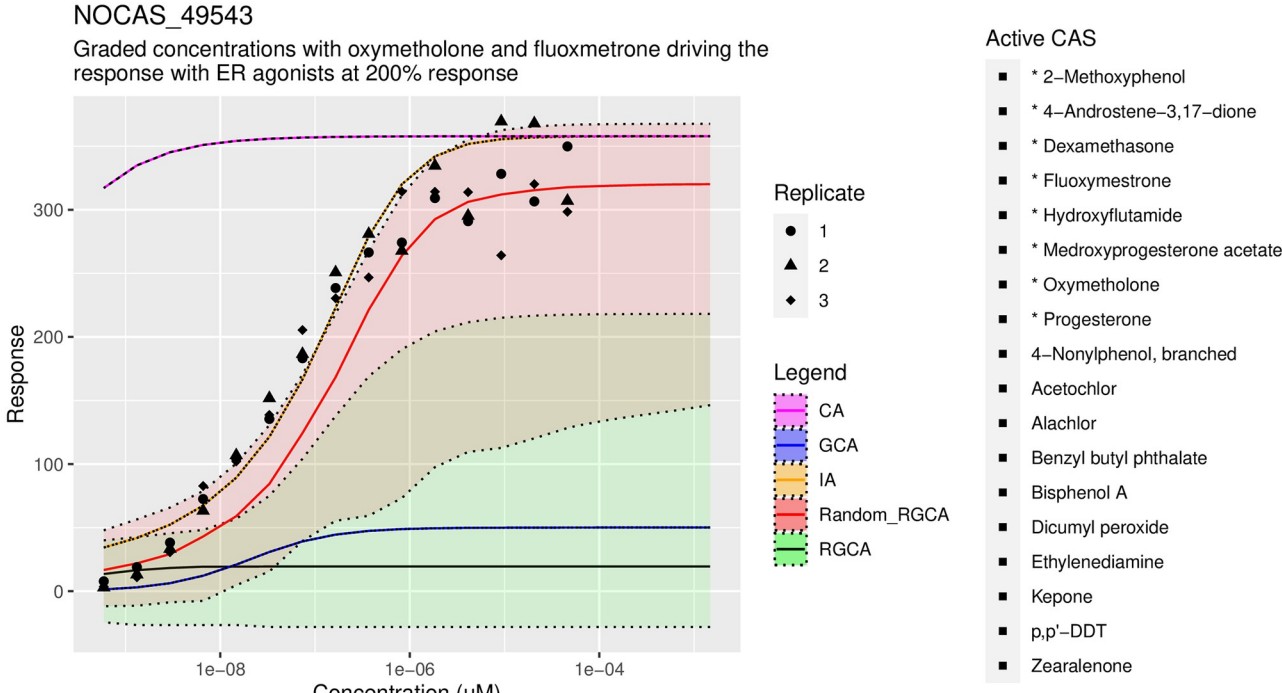

**Fig 6. Accurate predictions with RGCA.** A representative plot showing how mixtures are predicted for various concentrations and methods. All chemicals are present in this mixture, which is described as Mix 20 in the Supplement, S6 Table in S1 File. Our recommended method, RGCA with random clustering, is labeled as Random RGCA. IA, CA, and GCA are shown in yellow, pink, and blue lines, respectively. IA is better centered while our method covers most of the data with the credible interval. RGCA without clustering underestimates the truth due to the ER agonists, shown on the right without stars, that have small or negative sills.

In summary, our results suggest that the two-step method is more likely to cover the true mixture response when the truth is unknown.

## 4 Discussion

This work is motivated by the critical need for predicting a mixture response of an arbitrary set of chemicals from single chemical data only. First, we present Reflected Generalized Concentration Addition (RGCA), a piece-wise geometric extension of GCA that allows for non-unit slops. In contrast to existing methods, there are no restrictions on effect sizes or concentrations and we can use a 3-parameter Hill model to capture more of the structure in the individual dose response curves. Second, we can account for uncertainty in the data and predict a range of possible mixture responses. This demonstrates the importance of uncertainty quantification in mixtures toxicity prediction. Our method works best when there are many chemicals and fails (along with GCA and IA) in binary and small mixtures. We attribute this to the presence of synergy, which can have a strong effect when there are two chemicals of similar concentrations and can have a much weaker effect when there are many other chemicals to interfere in the interaction. This situation has been described as a "funnel effect" by [27].

The joint or two-step methods were generally the best approaches for the mixtures tested. As the mixtures involved more chemicals at significant doses, the IA model did quite well despite the assumption of a single mode of action with the assay tested. On one hand, this explains why our methods perform better than GCA: if IA is the truth and the two-step

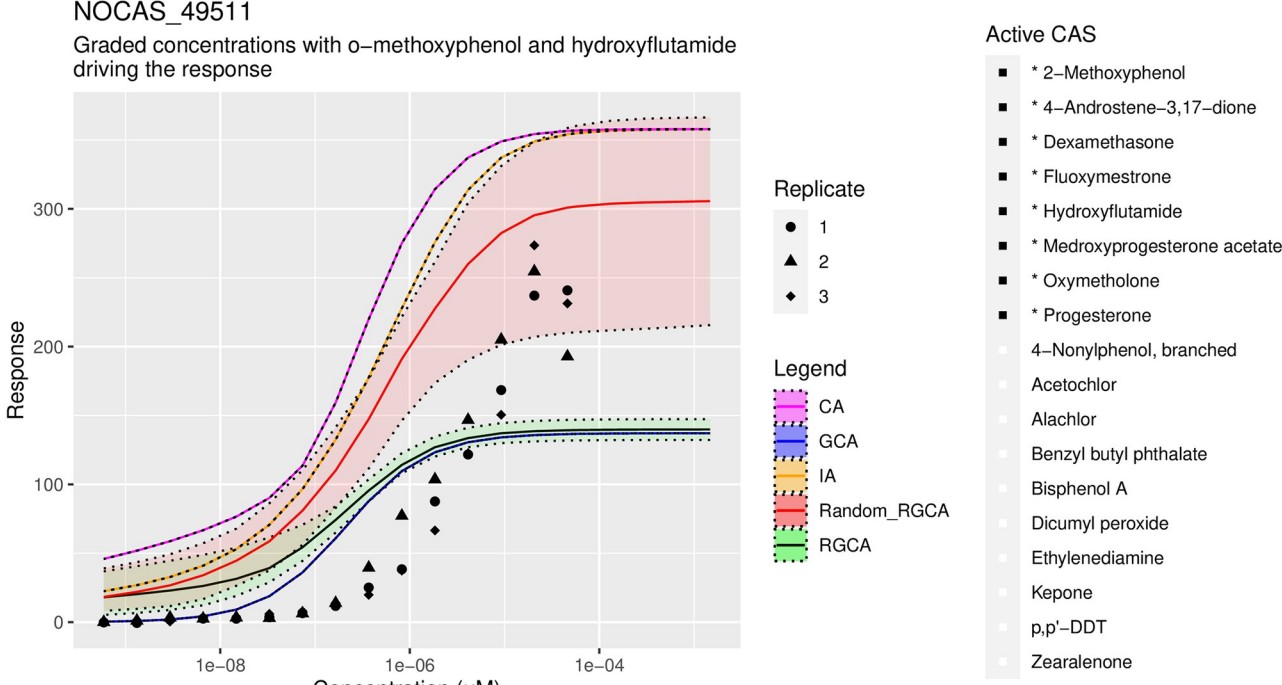

**Fig 7. Moderately accurate predictions with RGCA.** Only AR agonists are present, which is described as Mix 10 in the Supplement, S6 Table in S1 File. RGCA with random clustering is labeled as Random RGCA. IA, CA, and GCA are shown in yellow, pink, and blue lines, respectively. GCA fits best at very small concentrations but then underestimates the response. RGCA is similar to GCA at high concentrations but overestimates low doses because the estimated slopes are less than 1. Random RGCA along with CA and IA may be predicting the correct sill but observations at higher doses are not available.

method is interpolating between GCA and IA, then the two-step method will surely do better than GCA. On the other hand, perhaps the assumption of CA can be questioned: the experiment is cell based and the chemicals may be interfering with other processes that impinge on the AR receptor process, requiring IA rather than CA. In that case, we should be comparing to IA and will find that our method is not significantly better.

This leads to the main conclusion of our work: the two-step approach with RGCA and uncertainty quantification is most likely to cover the true mixture response when modes of action are not known. GCA and IA are likely to be the true boundary conditions for the prediction, but choosing just one carries a risk of large error. The two-step approach with random clusters or K-means clusters can reduce error and help the user determine which boundary is more relevant. The results of both simulation study and application support the use of the RGCA extension with random clustering when the slope parameters are far from 1 (ie outside of [0.5, 1.5]) and partial agonists are present. Even when CA is the correct model, forcing the slope parameters to be 1 when they are far from 1 can induce errors comparable to the interpolation error between a two-step method and GCA for moderate sizes mixtures.

## 4.1 Limitations and future directions

Our reflection argument allows for any positive slope value, but we found that there are numerical issues for combinations of non-unit slopes and negative sills. Fig 2 shows how multiple solutions arise and lead to distortions in the isobolograms that cannot be smoothed away. The problem lies in the optimization process, which can be susceptible to the local optima and

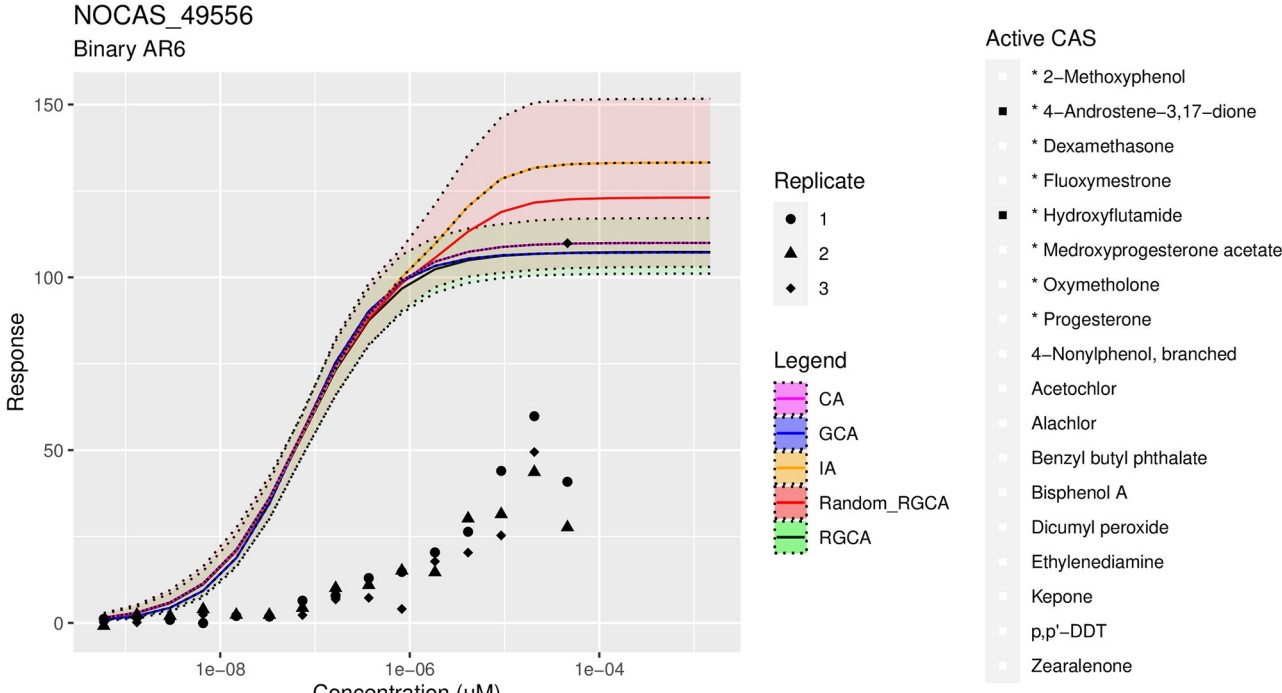

**Fig 8. Poor predictions with RGCA.** An example of poor predictive performance with a binary mixture. All methods fail to recover the true response. This mixture is exhibiting antagonism, which is not accounted for in any of the models.

very steep functions that result from non-unit slopes as shown in Fig 2. We describe a simple heuristic of always taking the positive solution when present, but it is not possible to specify which positive solution will be found when there are multiple solutions and the optimization may simply fail to find the solution because it was lost in a local optima. Investigating better numerical methods or heuristics is an area of future work. Methods for dealing with stiff differential equations show promise.

Although our approach demonstrates the applicability of the two-step approach, the question of finding a true or correct grouping that reflects the actual modes of action is not addressed. We explore the use of simple random or K-means clustering and find that it is often better than using a single cluster, but the predictions are never entirely correct. In short, the mixture problem isn't solved by interpolation. Furthermore, our methodology assumes a dose responses that is increasing, which is not directly applicable for interpolating biphasic responses that account for effects including cytotoxicity, such as those studied in [31].

Related to the concept of interpolation is whether or not chemicals with no observed effect should even be included in the calculation. Common sense suggests they should be left out, especially when using a method like GCA in which the no-effect chemical is treated as a partial agonist that strongly attenuates the predicted response. However, when an individual dose response shows no effect, it is not clear if the chemical is actually binding to the receptor with zero efficacy or not binding at all. If the chemical is binding to the receptor, it could be a competitive antagonist and must be included if possible in a mixture prediction. The Tox21 database contains such antagonist assays and also includes mixtures that include and exclude zero-effect chemicals; a preliminary study suggested inclusion of zero-effect chemicals, but such a study is beyond the scope of the present work.

The questions about mode-of-action and no-effect chemicals are side effects of the main limitation of our work: the exclusion of synergy. The challenge with synergy is the introduction of additional parameters controlling which chemicals synergize or antagonize with others. These are difficult to determine from individual dose response curves but possible with quantitative structure-activity relationships (QSAR), which would not require experimental data of the mixture. QSAR and docking or affinity scores may also help address the issue of no-effect chemicals by distinguishing between chemicals that do not occupy a receptor dock and those that do. With the inclusion of QSAR, it may be possible to use a true null model but identify slope parameters or cluster inclusion within our approach; this is useful because the chemical may have no effect but still synergize or antagonize another chemical. In contrast to IA and GCA, our approach is flexible enough to incorporate additional parameters or algorithmic steps utilizing QSAR data to account for synergy, which represents a promising future direction of research.

## Supporting information

**S1 File.**
(PDF)

## Acknowledgments

Thanks to Fred Parham and Mike DeVito for help with data.

## Author Contributions

**Conceptualization:** Kyle Messier.

**Formal analysis:** Daniel Zilber.

**Methodology:** Daniel Zilber, Kyle Messier.

**Software:** Daniel Zilber.

**Supervision:** Kyle Messier.

**Visualization:** Daniel Zilber.

**Writing – original draft:** Daniel Zilber.

**Writing – review & editing:** Kyle Messier.

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
