## [Decision Letter · Decision Letter 0]

22 Oct 2023

PONE-D-23-27967Reflected Generalized Concentration Addition and Bayesian Hierarchical Models to Improve Chemical Mixture PredictionPLOS ONE

Dear Dr. Messier,

Thank you for submitting your manuscript to PLOS ONE. After careful consideration, we feel that it has merit but does not fully meet PLOS ONE’s publication criteria as it currently stands. Therefore, we invite you to submit a revised version of the manuscript that addresses the points raised during the review process.

We look forward to receiving your revised manuscript.

Kind regards,

Y-h. Taguchi, Dr. Sci.

Academic Editor

PLOS ONE

Journal Requirements:

"Thanks to Fred Parham and Mike DeVito for help with data. This work is supported by the

444 National Institute of Environmental Health Sciences, Division of Translational Toxicology,

445 Division of Intramural Research, and the Spatiotemporal Exposures and Toxicology group

446 under project number ZIA ES103368-02."

"No, The funders had no role in study design, data collection and analysis, decision to publish, or preparation of the manuscript."

Reviewers' comments:

Reviewer's Responses to Questions

**Comments to the Author**

1. Is the manuscript technically sound, and do the data support the conclusions?

Reviewer #1: Yes

Reviewer #2: No

Reviewer #3: Partly

2. Has the statistical analysis been performed appropriately and rigorously? 

Reviewer #1: Yes

Reviewer #2: I Don't Know

Reviewer #3: Yes

3. Have the authors made all data underlying the findings in their manuscript fully available?

Reviewer #1: Yes

Reviewer #2: No

Reviewer #3: Yes

4. Is the manuscript presented in an intelligible fashion and written in standard English?

Reviewer #1: Yes

Reviewer #2: Yes

Reviewer #3: Yes

5. Review Comments to the Author

Reviewer #1: The development of joint toxicity modeling has been one of the challenges in the field, to which the authors of this work have made their own contribution. However, some issues still require further explanation or expansion of the data by the authors.

Lines 21-22: GCA is proposed to compensate for the shortcomings of the CA model, so it is suggested that the authors also need to provide the predictions of the CA model for comparison.

Lines 24-25: The presentation is too general and the predictive power of the developed models should be described quantitatively. For example, how many times higher than the predictive power of traditional models?

Lines 28-29: This problem is difficult to solve; after all, the models developed are also based on the prerequisite of non-interaction between the mixed components. And, existing models require a clear mode of toxic action.

2.1 section: Personally, I think that there is no theoretical basis for the integration of these two models into one. the GCA model is developed on the basis of the CA model on the premise that the mixed components should have the same modes of toxic action, whereas the IA model is applied on the premise that the mixed components have different modes of toxic action.

Figure 6: Too many constraints may limit the predictive efficiency of the model.

2.3 section: The incorporation of Bayesian Data Model may reduce the predictive efficiency of the model.

Result section: Increasing the results of the developed model in comparison with the CA model is more convincing.

Lines 427-441: In addition to considering interactions between hybrid components, the authors need to consider applying additional data sets to validate the models developed. In addition, the model involves too many parameters, which requires a larger amount of data for validation. From a risk assessment perspective, the practical applicability of the model still requires further consideration.

In summary, the authors of this study are still working hard to try to advance the mixture toxicity model. After considering the above reviewer' comments, the work is worthy of publication.

Reviewer #2: The manuscript refers to a specific mixture scenario where mixture components have differences in efficacy (ie, dose-response curve maxima) and, as consequence, the calculation of an expected mixture effect according to dose additivity is restricted. Solutions for this problem have been published previously but rely on narrow, usually unrealistic model assumptions (Howard &Webster) or provide a range of worst-case mixture predictions (Toxic unit extrapolation, Scholze et al). Here the manuscript suggests a mathematically-motivated more flexible approach. This is integrated in a chemical grouping approach where the authors make the assumption that the model parameter describing the steepness of a dose-response relationship is capable of showing a toxicological mode of action.

As much as I would like to see the use of novel mathematical & statistic methods in mixture toxicology, I have serious problems with the authors’ methods as they rest on assumptions which are not in line with current knowledge in mixture toxicology and pharmacology:

• The steepness of dose-response pattern is not indicative for the pharmacological/toxicological mode of action, a stubborn myth which experimental and theoretical toxicology have disproved now for decades. Compounds with nearly identical hazard profiles but different kinetic properties can result into data of very different dose-response shapes, compounds with different targets can produce nearly identical dose-response pattern, and this holds true for in vivo and in vitro study endpoints. So no, establishing cumulative mixture assessment groups according to the steepness of dose-response patterns is not an option anybody would suggest, and I cannot think about any application where a grouping of this specific model parameter under consideration of statistical variability would be useful.

• A receptor-based bioassay endpoint from eg the AR-luc assay is not capable of showing responses to more than one “mode of action”. In fact, most cell-based assay endpoints from the ToxCast library are annotated as MIE’s, and therefore considered as ideal for being in line with the pharmacological assumptions of Concentration addition. Also, factors like (i) nominal vs bioavailable (or target) concentrations, (ii) cytotoxicity at high concentrations, and (iii) lack of data support for weak assay responses and their impact on model complexity, can impact the curve estimate from an empirical dose-response model.

• Demonstrating by simulation that IA and CGA cannot describe a mixture scenario accurately that doesn’t fulfil the conceptual requirements of these two mixture models is trivial, so I don’t understand the meaning of paragraph 2.6, especially as it is promised to be a “validation”.

• Dose-response data from the AR-luc assay were used for 18 compounds, with 7 positively identified as AR antagonists and the remaining 11 known to be ER agonists. The authors say “there are 69 different mixtures” but without explaining the “where”, and it is unclear why and how these data sets were used for the simulations. For this endpoint only C(G)A is a reasonable mixture reference, and this has been demonstrated in numerous mixture studies. So why considering IA and RGCA+DP(I assume this refers to the proposed CGA/IA model)?

• Real data from experimental mixture studies are not used (or shown) to assess the approach, so I cannot see evidence in favour for this method.

• I cannot see how the extension of the CGA model by Hobster&Webster works, as Eq 6 refers only to β=1, ie the regression model used by Hobster&Webster. Where is the novelty?

My recommendations:

- The extension of the CGA model by Hobster&Webster deserves certainly its own publication, but here I would expect a better and more clear mathematical presentation, the (explicit or implicit) mathematical equations for the case β≠1 (Eq 6 refers only to β=1, or?), statements about the model domain (what are the limitations?), and a real mixture scenario to demonstrate its applicability (as in Howard&Webster or Scholze et al). The reader must be in the position not only to use the method but also to repeat the calculation for any example shown.

- If I got it right, the Bayesian aspect refers mainly to the between-study variability in the dose-response modelling, which is not novel (eg, the recommended approach by EFSA/Europe in the derivation of a regulatory benchmark dose). Here I would suggest to expand this to the proposed CGA model, expressing the statistical uncertainty of CGA prediction via a credibility interval is certainly something of great interest in the mixture field.

I would also strongly recommend to align with a mixture toxicologist.

Reviewer #3: In their manuscript "Reflected Concentration Addition to Improve Chemical Mixture Prediction" Daniel Zilber and Kyle Messier suggest an adopted approach to predict effects of chemical mixtures from observed effects of the mixture components.

In my view the approach is very interesting and plausible and can be a very valuable contribution for the field of mixture toxicology. The manuscript suffers in part from some ambiguities (see below) which should be addressed before publication. Overall, I would recommend considering the manuscript for publication after moderate revisions.

In my view, the following major topics should be addressed in a revision:

1) Incomplete and biphasic concentration responses in Tox21 Data:

An accurate mixture prediction depends a lot on accurate concentration response models for the mixture components. The authors partly address this issue by integrating a set of noise terms in the Hill function used. Yet, it did not become clear to me, how the authors deal with

a) biphasic responses (e.g. due to cytotox) as it can be seen for progesterone or hydroxyflutamide in Figure 2. I guess the highest 2 (?) concentrations are just not considered for the model, but it is not mentioned anywhere. It might also have a large effect on the overall outcome which concentrations are selected for modeling.

There are some studies on adopted CA for biphasic models (e.g. http://www.nature.com/articles/srep17200) and it would be interesting to see how these compare to the proposed RGCA model. Since this could be quite a lot of additional work and thus not feasible, I think it would be really good to include this topic in the discussion.

b) "incomplete" concentration response/unknown maximum effect: The authors thoroughly address the topic of partial agonists/variable maximum effect. Especially for High-Throughout-Data like Tox21, we regularly have the issue, that effects do not reach a plateau, so we only have an incomplete curve and a maximum effect cannot be identified. It gets even worse for the case of cytotox, where the maximum effect is even more obscured. This does not matter for calculating Benchmark Concentrations, but is a very important limitation in mixture modeling. It would be very helpful if the authors make more explicit how the maximum effect is modelled. If feasible, they could make a very strong point, when they could test the sensitivity of different alpha ranges on mixture prediction. I am aware, that this might be too much outside the scope of the study, but it should made explicit in the methods and included as limitation in the discussion at least.

2) Benefits of the Bayesian Hierarchical Models in Mixture Modelling: As the authors point out, earlier mixture studies have shown that Observed Mixture Responses often end up somewhere between IA and CA. When reading the manuscript I wondered, if one could arrive at the same quality of prediction as with the Bayesian Clustering by just randomly assigning the clusters (maybe still based on slope, but e.g. a random cluster number). Would you dare to try it out? ;-)

3) I found the manuscript very interesting and inspiring to read, but I got the feeling that there should be some switching between SI and manuscript paragraphs. In some parts, established methods (like DPMM) were explained in detail - which is a nice service to the reader - but distracts a bit from the main adoptions and findings of this study. On the other hand, important information about the methods (e.g. how and why did we chose priors for single chemical modeling) are only implicitly mentioned or buried in the supplement. Maybe this is also a matter of taste, but I just want to encourage the authors to check theirs manuscript for this once more. Also make sure, that references to the SI are clear and easy to follow.

Some more minor issues and questions in the following:

Abstract:

l. 13: "This is purely predictive..." This only becomes clear in the context of approaches that use mixture observations for modelling. Maybe reframe, e.g. "This predicts mixture responses based on observed responses of mixture components"...

l. 25: "significantly improved...": In my view, this does not go in line whith your discussion, where you state, that CRPS is your recommended metric for drawing conclusions (l. 389) and CRPS showed "marginally better" results (l. 386).

l. 28: "Lastly..." I would recommend to leave out this sentence, since this is not scope of the study

Introduction:

ll. 34ff: I share your concerns regarding "forever" chemicals, but was not convinced, that this group specifically adds to the mixture problem...

ll. 39-51: I really liked this introduction into the world of mixture prediction!

ll. 52ff: Maybe consider to cite the original publications for CA and IA?

l. 58: Could be misleading to talk about "adding" effects. Maybe reframe to "...so a series of chemical proportional effects can be multiplied..."?

ll. 60ff.: It took me a while to understand which probem(s) you are actually adressing with your study, maybe because there are these two elements of a) clustering similar mode-of-action and b) reflected gca, which both adress different types of problems. Maybe this could be made even more explicit in the introduction

l. 74: This statement did not became clear to me, maybe you could elaborate a bit more on this

Material and Methods:

"Algorithm 1": This did not help me too much for understanding the approach. To be able to grasp this more intuitively, maybe a flowchart would be an option. Otherwise, it would at least help to explain/define all mentioned parameters and abbreviations, be a bit more consistent in the structure (e.g. "Step 1...Step 2... Step 3a...Step3b" in the "Result" and Approach). I could not find the referenced Equation 7 in the Supplement (Equations are not numbered there).

Equation 3: Please also define/explain parameters of this equation. How is alpha_max different from alpha (equation 1)? How do you deal with different alpha_max values in the IA part?

Equation 6: I got a bit confused here. You are talking about a general form of inverse for beta not 1, but in Equation 6 you state beta = 1?

l. 162: "iid Gaussian noise" - Typo?

l. 170: Please state and justify all priors here or in the supplement.

l. 180: Please explain how the data was filtered before modeling (see above).

Thanks for supplying your R code on Github. I saw a summary Rmd file there. Maybe it would be worth considering to include the rendered pdf as a supplement to the manuscript?

ll. 197-210: Nice description of the process, but maybe consider to move to supplements

l. 235: Very important point, that the slopes within a cluster are identical. Do you consider this when sampling the remaining parameters, that you only sample from the chains with the corresponding fixed slope?

Is the RGCA approach in general also applicable for differing slopes?

ll. 252-262: consider to move to supplements

l. 289-290: It did not became clear to me, how you arrived at "realistic" Hill parameters, and what "feasible clustering conditionals" means. Please also state how many different mixtures you simulate, with how many chemicals included and how the mixture compositions are defined.

Results

l. 317: It would help to reiterate your assumptions here

Figure 3: Please state n. Would it be feasible to simulate larger mixtures?

l. 336: This is a very interesting finding. It would be nice, if you could plot the fits and the mentioned extrapolations, so one could judge how they differ and which would be more plausible. Include here or in the supplement.

l. 347: "below" Figure 4?

l. 384: Maybe you could draw a dot/ or boxplot showing CRPS score against the number of mixture components and different colors/shapes for the different prediction methods?

Figure 6: What are the large circles? replicate 1? It is confusing that they are larger than Replicate 2 and 3 and not mentioned in the legend. In my opinion more fits would deserve to be included in the main part of the manuscript. It would also be interesting to include the CRPS score for the different models in the plot so one could get a feeling for what the numbers mean. Consider to include at least a plot with a very low score/ a medium score and a high score...Include all the remaining fits in the supplement.

Discussion:

l. 380: "Fails with binary and small mixtures": When looking at Supplemental Figure 5, Binary AR2 and AR7, I get the impression, that your RGD approach actually performs not totally bad, since the measured responses fall in the (quite broad) credible interval? What did not become clear to me, why the credible interval is so wide for RGD exclusively, because the clustering cannot be so variable for 2 substances, right? Could you elaborate on this?

l. 422: Workaround would then be back to GCA?

ll. 427ff: Can you relate to any exisiting literature on this topic?

6. PLOS authors have the option to publish the peer review history of their article (what does this mean?). If published, this will include your full peer review and any attached files.

Reviewer #1: No

Reviewer #2: No

Reviewer #3: No

---

## [Author Response · Author response to Decision Letter 0]

27 Dec 2023

See attached response_to_reviewers pdf document.

---

## [Decision Letter · Decision Letter 1]

30 Jan 2024

Reflected Generalized Concentration Addition and Bayesian Hierarchical Models to Improve Chemical Mixture Prediction

PONE-D-23-27967R1

Dear Dr. Messier,

We’re pleased to inform you that your manuscript has been judged scientifically suitable for publication and will be formally accepted for publication once it meets all outstanding technical requirements.

Kind regards,

Y-h. Taguchi, Dr. Sci.

Academic Editor

PLOS ONE

Additional Editor Comments (optional):

Reviewers' comments:

Reviewer's Responses to Questions

**Comments to the Author**

1. If the authors have adequately addressed your comments raised in a previous round of review and you feel that this manuscript is now acceptable for publication, you may indicate that here to bypass the “Comments to the Author” section, enter your conflict of interest statement in the “Confidential to Editor” section, and submit your "Accept" recommendation.

Reviewer #1: All comments have been addressed

2. Is the manuscript technically sound, and do the data support the conclusions?

Reviewer #1: Yes

3. Has the statistical analysis been performed appropriately and rigorously? 

Reviewer #1: Yes

4. Have the authors made all data underlying the findings in their manuscript fully available?

Reviewer #1: Yes

5. Is the manuscript presented in an intelligible fashion and written in standard English?

Reviewer #1: Yes

6. Review Comments to the Author

Reviewer #1: (No Response)

7. PLOS authors have the option to publish the peer review history of their article (what does this mean?). If published, this will include your full peer review and any attached files.

Reviewer #1: No

---

## [Editor Report · Acceptance letter]

19 Mar 2024

PONE-D-23-27967R1 

PLOS ONE

Dear Dr. Messier, 

I'm pleased to inform you that your manuscript has been deemed suitable for publication in PLOS ONE. Congratulations! Your manuscript is now being handed over to our production team.

Kind regards, 

on behalf of

Professor Y-h. Taguchi 

Academic Editor

PLOS ONE